# Immuno-Oncological Biomarkers for Squamous Cell Cancer of the Head and Neck: Current State of the Art and Future Perspectives

**DOI:** 10.3390/cancers13071714

**Published:** 2021-04-04

**Authors:** Stijn J. De Keukeleire, Tijl Vermassen, Elien Hilgert, David Creytens, Liesbeth Ferdinande, Sylvie Rottey

**Affiliations:** 1Department of Medical Oncology, University Hospital Ghent, 9000 Ghent, Belgium; Tijl.vermassen@uzgent.be (T.V.); sylvie.rottey@ugent.be (S.R.); 2Cancer Research Institute Ghent (CRIG), 9000 Ghent, Belgium; Elien.Hilgert@ugent.be (E.H.); David.creytens@uzgent.be (D.C.); liesbeth.ferdinande@uzgent.be (L.F.); 3Drug Research Unit Ghent, University Hospital Ghent, 9000 Ghent, Belgium; 4Centre for Medical Genetics Ghent (CMGG), University Hospital Ghent, 9000 Ghent, Belgium; 5Department of Pathology, University Hospital Ghent, 9000 Ghent, Belgium

**Keywords:** squamous cell cancer of the head and neck, immunology, immune checkpoint inhibitors, biomarkers, tumor microenvironment, human papilloma virus, immune infiltrate, genetics, microbiome

## Abstract

**Simple Summary:**

Squamous cell cancer of the head and neck is a common malignancy with poor prognosis. Despite the success of PD-L1 expression, the landscape of diagnostic, prognostic, and predictive biomarkers has delivered limited contributions to the clinic in the last decade. The dissection of the immunological landscape through investigation of the immune infiltrate, blood-based biomarkers, and genetic profiling has shown substantial scientific potential but all are yet to be validated. Further exploration is warranted though implementation of biomarkers. This should ideally be performed through prospective studies using standardized methods with harmonization of technical requirements. This review serves as a comprehensive overview for state-of-the-art knowledge and biomarkers in squamous cell cancer of the head and neck (SCCHN) that may prove their worth in future clinical practice.

**Abstract:**

The era of immune checkpoint inhibitors has altered the therapeutic landscape in squamous cell cancer of the head and neck (SCCHN). Our knowledge about the tumor microenvironment has fueled the research in SCCHN, leading to several well-known and less-known prognostic and predictive biomarkers. The clinical staging, p16/HPV status, and PD-L1 expression are currently the main tools for assessing the patients’ diagnosis and prognosis. However, several novel biomarkers have been thoroughly investigated, some reaching actual significant clinical contributions. The untangling of the immune infiltrate with the subtyping of tissue-associated tumor infiltrating lymphocytes, tumor-associated macrophages, and circulating blood-based biomarkers are an interesting avenue to be further explored and prospectively assessed. Although PD-L1 expression remains the most important response predictor for immune checkpoint inhibitors, several flaws impede proper assessment such as technical issues, different scoring protocol, and intra-, inter-, and temporal heterogeneity. In addition, the construction of an immune-related gene panel has been proposed as a prognostic and predictive stratification but lacks consensus. Recently, the role of microbioma have also been explored regarding its systemic and antitumor immunity. This review gives a comprehensive overview of the aforementioned topics in SCCHN. To this end, the integration of these clinically advantageous biomarkers via construction of an immunogram or nomogram could be an invaluable tool for SCCHN in future prospects.

## 1. Introduction

Squamous cell carcinoma of the Head and Neck (SCCHN) globally affects more than 750,000 new cases (estimated 4 percent of the global cancer incidence) and 360,000 deaths annually [1,2]. In Europe (EU-27), approximately 70,000 new cases and around 40,000 deaths have been registered in 2020 [3,4,5]. The majority of patients (50–60%) presents with loco-regional advanced stage III–IV disease. Major risk factors that induce carcinogenesis in the head and neck region are excessive alcohol and tobacco abuse, and account for over 80% of all diagnosed SCCHN [6,7]. Human Papillomavirus (HPV) status also plays an important role in the development of SCCHN, and HPV positivity is observed in around 20% of all SCCHN cases, of which nearly 70% is situated in oropharyngeal squamous cell carcinoma (OPSCC). Several studies have correlated HPV ^+^ cancer, especially in the oropharyngeal region, to improved therapy response and increased 5-year overall survival (OS; 83% versus 37% HPV^+^ versus HPV^−^ cancer) [8,9]. Patients who are diagnosed with recurrent or metastasized (R/M) SCCHN have a poor prognosis with only a handful of systemic therapeutic options, mostly platin-based chemotherapies [10,11].

Unmistakable evidence has been gathered in recent years regarding the pivotal role of the immune system in cancer development and progression, which is commonly referred to as the tumor microenvironment or TME. General concordance exists that a deficient immune-surveillance is largely induced by neoplastic cells. Plausible theories are the lack of proper antigen recognition and/or presentation, enhanced production of immunosuppressive mediators, e.g., cytokines, and an overall reorganization of the cellular constituents to sustain tumoral formation [12,13,14]. Immunotherapy based on immune checkpoint inhibition (ICI) made its entry in SCCHN and has recently been implemented as a first and/or second line treatment in R/M SCCHN, as proven effective in recent trials in SCCHN with overall response rate (ORR) reaching up to 18.2% [12,15,16,17]. ICI responsive patients do have a longer duration of response and a better safety profile compared to patients who receive standard treatment [18]. Therefore, the implementation of ICI has been considered a successful step in the treatment of SCCHN. However, the low response to ICI can partially be explained due to the heterogenous nature of SCCHN in regards to its genetic, molecular, and immunological profile. The precise mechanisms that induce immune escape remain undefined. 

It is without question that TME has established a dominant role in the oncological treatment landscape [19,20,21]. As ICI have been integrated in daily oncological practice, the necessity for further diagnostic and therapeutic stratification is of vital importance [7]. To address this issue, several attempts have been made to dissect the immunological signature of SCCHN via the exploration and identification of feasible, robust biomarkers. The common definition states as a biomarker being a measurable characteristic of normal biological processes, pathological processes, or responses to certain exposures or therapeutic interventions [22]. For cancer research, biomarkers can be grossly divided into (1) diagnostic markers, aiding in the (early) detection and diagnosis of the disease, (2) prognostic markers, delivering information about the patients’ health outcome, and (3) predictive markers, informing about the response probability of a certain therapeutic intervention. Before a biomarker assay can undergo the steps of discovery, verification, and clinical validation, an extensive collection of high-quality patient samples or biospecimen should be assembled. Various types of biospecimen exist, though they are mostly comprised of tissue samples (fresh, frozen, or Formalin Fixed Paraffin Embedded [FFPE]), blood and blood derivates (whole blood, plasma, serum, peripheral blood mononuclear cells (PBMC)…), biofluids (urine, synovial, and cerebrospinal fluid…) and other derivatives, molecules, or non-specified formats (DNA, RNA, proteins, stained tissue slides, swabs, etc.). These biospecimen are to be collected from patients retrospectively or prospectively. An ideal biomarker should be feasible, robust, cost-effective, and of significant clinical value. The application and interpretation should be widely available. In case of prognostic and predictive biomarkers, especially in cancer research, these should be defined on the basis of clinically valid surrogate endpoints such as response rate or survival rates [22,23,24,25].

In this review, we will comprehensively elaborate on some general aspects of the TME while discussing several crucial diagnostic, prognostic, and/or predictive biological markers that have been extensively explored in SCCHN. Finally, some suggestions and future perspectives will be presented that may inspire researchers and clinicians to integrate the immunological profile in the diagnostic and therapeutic landscape of SCCHN.

## 2. General Concepts of Tumor Immunology

The tumor microenvironment or TME is a complex, dynamic environment that is shaped during cancer progression and may dictate tumor survival and growth by a continuous, bidirectional interaction between tumor and host cells. The eventual goal is to sustain paracrine and juxtacrine (contact-depending signaling) growth factor, nutrient- and oxygen supply, and the neutralization of an anti-neoplastic immune response, assuring tumor survival and progression. It is commonly built of neoplastic cells, supportive tissue (extracellular matrix) and cells (fibroblasts), and an immune infiltrate consisting of a grand variety of immune cells, such as neutrophils, natural killer (NK) cells, tumor associated macrophages (TAMs) myeloid derived stem cells (MDSC), dendritic cells, and several subsets of lymphocytes such as CD4^+^ T helper cells, CD8^+^ cytotoxic T cells, and regulatory CD4^+^Fox(forkhead box)P3^+^ Tcells (Treg). The immunosuppressive cell recruitment is accompanied by increased secretion of immunosuppressive cytokines (tumor necrosis factor alfa (TNFα), tumor growth factor beta (TGFβ), interleukin (IL)-10, interferon gamma (IFNγ), etc.), enzymes (arginase 1 (Arg1), inducible nitric oxide synthase (iNOS), reactive oxygen species (ROS), etc.) and upregulated expression of several surface molecules that enhance immunosuppression [26]. On this matter, expression of immune checkpoint molecules (e.g., programmed death ligand 1 and 2 (PD-L1/PD-L2), cytoxic T-lymphocyte associated protein 4 (CTLA-4), indoleamine 2,3-dioxygenase (IDO), T cell immunoglobulin and mucin domain-containing protein 3 (TIM-3), Killer-cell immunoglobulin-like receptors (KIR), T cell immunoreceptor with Ig and ITIM domains (TIGIT)) on tumor cells, antigen presenting cells (APCs), and several immune cells have led to additional proof that tumors up-regulate an immunosuppressive environment as they function as negative regulators of the T cell immune function [12]. An important finding was discovery of the positive-feedback loop between Treg cells and TAMs which is essential to maintain or promote immunosuppression within the TME (Figure 1) [27].

Furthermore, the PD-1/PD-L1 axis is an essential mechanism for maintaining peripheral tolerance and restraining over-activity of self-reactive immune cells from causing harm by inducing T cell anergy, exhaustion, and apoptosis. Alternatively, blockade of the CTLA-4/B7 axis via CTLA-4 antibodies (Abs) has been correlated with expansion of the T cell antigen-recognition repertoire, restraining the over-activity of the immune system by inducing T cell anergy and/or apoptosis, thus maintaining self-tolerance [7,12,17,28,29,30,31]. CTLA-4 and PD-1 may be both expressed by CD4^+^/FoxP3^−^ T helper cells, (CD4^+^/FoxP3^+^ regulatory Tcells (Tregs)) and CD8^+^ cytotoxic T cells, while PD-1 is also present on macrophages, B-cells, NK cells, myeloid derived stem cells (MDSCs), and other APCs. Subsequently, tumor cells may overexpress the immune checkpoint-associated ligands or components (PD-L1 and B7-1 or B7-2), thus avoiding potential anti-tumor immune responses (Figure 1). Therefore, inhibition of these proteins could lead to the recovery of the immune system, enabling the cytotoxic properties of f.i. NK cells and lymphocytes, as seen with the current ICI immunotherapeutic Abs, while increasing the risk for presence of T cell clones with potentially auto-reactive characteristics [15,16,32,33,34]. 

As mentioned earlier, a continuous bidirectional interaction between immune cells and neoplastic cells takes place: during progression of carcinogenesis, cancer cells resist elimination via the immune editing process. This concept in tumor biology is based on the principles of immunosurveillance, which has been profoundly described by several authors the last few decades [26,31,35,36]. As the processes of immune-editing leads to potential selection of tumor cells, some may become less immunogenic. This is mostly achieved via acquisition of mutations and/or loss of expression in the antigen processing machinery or human leukocyte antigen (HLA) encoding genes, reducing antigen expression through major histocompatibility complex (MHC) molecules, hence resisting T cell recognition and elimination [7,26]. Indeed, data from the cancer genome atlas (TCGA) has indicated that these mutations have been observed on large scale in SCCHN. Furthermore, it seems this process does not only occur during tumor progression, but can also be reinitiated while under treatment with (ICI-based) immunotherapy, thus inducing potential resistance to these agents [36]. It is without question that the TME is a complex matter of which our current knowledge remains insufficient. Understandably, current clinical trials now have their focus on combinational strategies for targeting multiple elements of the TME, but with a main focus on implementation of ICI. Nonetheless, extensive knowledge regarding the TME in SCCHN can only be beneficial in our search for raising the chances of gaining an anti-tumor response with immunomodulating agents [37].

## 3. The TME in SCCHN

SCCHN is a heterogeneous disease with a variety of genetic alterations and is generally associated with an immunosuppressive environment. Overall, the latter is known to restrict survival in cancer patients which could partially explain the relatively poor survival statistics of SCCHN. OPSCC has the most favorable prognosis of SCCHN subsites, which may be attributed to its typical inflammatory environment and high immune infiltration [20]. Literature grossly divides the TME of SCCHN in two immunogenic phenotypes. Firstly, the inflamed, virus-driven phenotype, which is characterized by high immune-infiltration, increased radio-and chemotherapy sensitivity, and prolonged survival. In fact, according to Mandal et al. [28], these subtypes have one of the highest rates of immune cell infiltration in solid cancers, of which the highest Treg and (CD56^dim^) NK cell infiltration [38]. Next, the alcohol/tobacco-induced, immunosuppressive TME has overall low immune-infiltration and lower survival rate. It seems that this subgroup of patients have a suppressed CD8 mediated anti-cancer response, objectified by a lower IFNγ-signaling and reduced concentration of immune effectors granzyme and perforin [39]. Furthermore, this is complicated by the “mutational smoking signature,” originating from Alexandrov et al. [40], which may exert immunomodulatory changes to both phenotypes. This has also been described in non-small cell lung cancer, as patients with smoking signatures seem to induce pro-inflammatory effects on the TME and benefit from a higher response rate to ICI [39,41]. This topic will not be discussed in further detail as this is beyond the scope of this review.

Nevertheless, a diagnostic and prognostic assessment in SCCHN should not be restricted to the tumor’s clinical stage, differentiation grade, or P16/HPV status [7,42,43,44]. In the following sections, several relevant topics connected to the immunological phenotype in SCCHN are described that may provide valuable diagnostic, predictive, and prognostic information.

This manuscript will be unable to cover all multi-omic biomarkers within SCCHN, but gives an overview of several state-of-the-art biological factors that have been extensively explored, assessed, or (nearly) implemented in clinical practice. Several of these are or could be valuable for clinicians when utilizing immunotherapeutic agents. We will cover several tissue-based and genetic biomarkers and explore the role of circulating blood cells while also giving brief consideration to the oral microbiome in SCCHN.

### 3.1. Tissue-Based Biomarkers

#### 3.1.1. HPV/P16 Status

It has been sufficiently demonstrated that patients presenting with HPV^+^ OPSCC show improved response to treatment and have a better disease-specific as well as disease-free survival (DFS). High-risk HPV is associated with malignancy, mostly in the cervix, vulva, vagina, penis, anus, rectum, and oropharynx (including the base of tongue and tonsil) [35]. The WHO currently identifies 12 high-risk cancer-causing HPV strains (types 16, 18, 31, 33, 35, 39, 45, 51, 52, 56, 58, and 59). HPV 16 has been acknowledged as the causative strain for development of OPSCC [45,46]. HPV types 31, 33, 45, 52, and 58, combined, are linked to approximately 10% of all HPV^+^ cancers. An indicator that is typically associated with HPV infection in OPSCC is the immunohistochemical overexpression of the P16 protein (CDKN2A). This tumor suppressive protein directly and indirectly regulates RB (Retinoblastoma protein) and P53 function, two crucial elements involved in normal cell homeostasis that can be affected by HPV-related oncoproteins E6 and E7, which promote degradation of RB and P53 [47]. Detection of HPV is achieved through polymerase chain reaction, in situ hybridization, or immunohistochemical techniques using P16 overexpression. The latter serves as a surrogate marker for SCCHN, especially OPSCC with sensitivity and specificity reaching up to 94 and 83%, respectively [45,48,49]. In non-oropharyngheal SCCHN, however, the HPV-positivity rate is lower and P16 positivity is correlated to lower specificity. In addition, it seems that oral squamous cell cancer (OSCC) does frequently over-express P16, but is only rarely HPV-driven [50,51]. Consequently, this may lead to discordant results when P16^+^/HPV^−^ are incorporated as truly HPV-driven SCCHN in survival analyses [50]. 

P16/HPV positivity has a considerable effect on the TME of SCCHN, as HPV positivity may be correlated with expression of foreign virus-related antigens, thus inducing a higher inflammatory response compared to P16/HPV negative SCCHN. Indeed, P16 and/or HPV positivity in OPSCC is correlated with better response to radio- and chemotherapy, a more favorable OS and lower likelihood of relapse in comparison to OPSCC with negative P16 and/or HPV status. Nonetheless, the relationship between P16/HPV status and lymphocyte infiltration remains controversial, as patients with high lymphocyte infiltration have better OS and DFS (especially in OPSCC), but this was considered independent of HPV status [8,47,52,53]. However, a higher viral load in HPV^+^ OPSCC is correlated with increased immune infiltration, implicating that a HPV^+^ status does attribute to but is not solely responsible for an induced higher local tumor inflammation [46]. Furthermore, a research paper from Lechner et al. [54] prospectively investigated primary tumoral tissue and blood (PMBC) of treatment naïve SCCHN patients (n = 34) and control patients (n = 15), observing no significant alterations in tissue-related T cell subsets when comparing HPV^+^ and HPV^−^ SCCHN

It is indisputable that HPV^+^ SCCHN has a better prognosis than HPV^−^ strains. This may be explained by the presence of an intact (non-mutated) *P53* gene in HPV^+^ SCCHN, rendering them more vulnerable to therapies than HPV^−^ SCCHN, which are known for their higher mutational burden [55,56,57]. As mentioned earlier, the recent implementation of ICI have increased the OS of therapy responding patients with R/M SCCHN significantly. Several ICI-based clinical trials have investigated if the overall response rate (ORR) is altered regarding P16 and/or HPV status. The phase Ib KEYNOTE 012 trial (n = 60) and phase II KEYNOTE 055 trial (n = 171) treated R/M SCCHN patients with the ICI pembrolizumab and noted a 22% and 32% ORR in HPV^+^ SCCHN compared to 16% and 14% in HPV^−^ SCCHN, respectively [58,59]. In a multicentre phase I/II study from Segal et al. [60], 62 R/M SCCHN patients were treated with the ICI durvalumab, of which 40.2% were HPV^+^. Remarkably, these patients had worse ORR than HPV^−^ SCCHN (0% versus 8%), though cohort size was rather limited (n = 50). Regarding survival, the CHECKMATE 141 was a phase III trial that enrolled 361 patients with platinum-resistant R/M SCCHN who were treated with the ICI nivolumab or with the investigator’s choice at a 2:1 ratio. In the nivolumab-treated cohort, patients with P16^+^ tumors had significantly higher OS than P16^−^ tumors (9.1 versus 7.5 months) [60]. Future prospective trials should further investigate the relationship between HPV/P16 status and PD-L1 expression, and if HPV status affects ORR or prognosis in ICI-treated SCCHN patients (cfr. 3.1.5) [61].

Nevertheless, HPV status and/or its surrogate marker, P16, remain of indisputable value during the diagnostic process of SCCHN. However, the relationship between HPV-status and the TME of SCCHN remains somewhat vague. Although HPV^+^ SCCHN seems to be significantly higher infiltrated by tumor infiltrating lymphocytes (TILs) than HPV^−^ SCCHN, no significant differences in TIL subsets have been observed. Further investigation is nonetheless required in correlating P16/HPV status to the TME [53].

#### 3.1.2. Tumor Immune Infiltration: Subtyping and Quantification

##### Tumor Infiltrating Lymphocytes (TILs)

TILs have been thoroughly investigated and acknowledged as a key part of the immune infiltrate and include NK cells, γδ T cells, NKT cells, CD4^+^ T cells, CD8^+^ T cells, and B cells. There is global consensus that mainly TILs are deregulated regarding number and functionality in SCCHN. The identification and quantification of several subsets of TILs in the TME has been thoroughly examined in SCCHN. The most investigated and clinically relevant subsets are CD3^+^, CD4^+^, and CD8^+^ T cells, visualized on tumoral tissue sections using immunohistochemistry (IHC) (Figure 2). 

CD3 is a pan-T cell marker functioning as a co-receptor for the T cell receptor, which is required for T cell activation. CD3^+^ infiltration can be deducted as a general marker for T cell infiltration, and several reports have correlated it with beneficial clinical outcome in SCCHN in comparison to low CD3^+^ infiltrated tumors by IHC-based staining and semi-quantification (Figure 2C,D) [62,63,64]. This again was contradicted by Lechner et al. [54] as no OS difference was found in high CD3^+^ infiltrated SCCHN tissue, not in the primary tumor nor in metastatic lesions.

The CD4 glycoprotein is a surface immunoglobulin (IgD) expressed on T helper strains while functioning as a co-receptor for the MHC class II complex. The prognostic value of CD4^+^ T cells remains questionable. A well-known subpopulation are CD4^+^ FoxP3^+^ T cells or Tregs which are associated with hosting an immunosuppressive environment, promotion of tumor survival and progression [65]. However, several retrospective studies evaluated the infiltration rate of CD4^+^ FoxP3^+^ on mainly OPSCC. FFPE tissue slides were stained by IHC and analyzed by conventional pathological quantification or digital image analysis. All articles concurred Treg cell density did not affect clinical outcome in OPSCC [66,67,68].

The cytotoxic T cells are a subpopulation of T cells that act as suppressors of tumor growth. By means of IFNγ production, expression of MHC class I tumor-related antigens is upregulated, allowing swift recognition and elimination by production of cytotoxic granzymes and perforin. These T cells are typically identified by expression of the CD8 membrane glycoprotein [69]. To this end, several articles reported increased infiltration of CD8^+^ lymphocytes by IHC retrospectively performed on tissue slides revealed a significantly better prognosis in SCCHN [52,63,66,67,70,71,72]. Studies from our research group revealed an increased CD8 expression in the immune infiltrate to be an independent prognostic variable in OPSCC, while CD3, and CD4-FoxP3 expression were not correlated to survival. Furthermore, these results were independent of P16/HPV-status [73]. Finally, a systematic review and meta-analysis from De Ruiter et al. [74] confirmed elevated CD3^+^ and CD8^+^ infiltration can be correlated with better prognosis in SCCHN, independent of HPV status.

Current findings regarding the prognostic value of TILs in SCCHN seem to be discordant. The factors that might enhance these differences have been elaborated in a previous review provided by our research group and are briefly summarized here, namely (1) biological heterogeneity: HPV status, topical heterogeneity; (2) technical factors: type of tissue specimen (biopsy/resection/metastasis) or employed antibody for IHC; and (3) the lacking of a standardized method of scoring TILs, the latter being a recurrent pitfall that impedes TILs from being implemented as additional diagnostic or prognostic markers. Several attempts have been made to develop standardized methods for TIL assessment. For instance, the Immunoscore is a method for (semi-) quantification of CD3^+^ and CD8^+^ infiltrating T cells introduced in renal cell carcinoma and colorectal cancer but has not been utilized in SCCHN as a potential diagnostic or prognostic biomarker [54,75,76,77,78]. Recently, the international immunology biomarker working group (IBWG) has designed a guideline-based protocol to assess (stromal and intratumoral) TILs on single slide haematoxilin-eosin (HE) stained sections in several types of solid carcinoma, including SCCHN (Figure 2A,B) [79,80,81]. Our research group has investigated current methodology in OPSCC in a retrospective fashion, showing a high amount of stromal TILs was an independent prognostic factor as patients with a high amount of stromal TILs has better OS compared to patients with low or absent stromal TILs, independent of P16 status (unpublished data). 

In conclusion, TILs remain an important aspect in the TME and should not be neglected when assessing SCCHN as they may conceal interesting prognostic information. Several TIL subsets have been thoroughly investigated by different research groups, and although TIL subtyping and quantification show promising potential as biomarkers, there is a lack of prospective trials to validate these findings.

##### Tumor-Associated Macrophages (TAMs)

Another strain of infiltrating immune cells are macrophages. These are recruited from the bone marrow as peripheral monocytes or originate from TME-attracted MDSC and polarize into two different macrophage phenotypes, M1 and M2 TAMs, depending on the received stimuli from the TME. M1 macrophages, which will primarily develop in presence of IFNγ, are acknowledged as potent effector cells for eliminating tumor cells by production of several pro-inflammatory cytokines and activating Th1 cells, thus inhibiting tumor progression [53,82,83]. More specifically, they induce activation of CD8^+^ cytotoxic cells and differentiation of naïve CD4^+^ T cells into Th1 effector cells [53]. Activated M1 macrophages can be distinguished by general expression of surface proteins HLA-DR and CD80/86, but numerous others have also been described (CD64, CD16, CD120b, TLR2, and SLAMF7 etc.) The activated M2 macrophages on the other hand are considered to be predominantly tumor associated, as they are characterized by the ability to produce anti-inflammatory cytokines (e.g., IL-10, TGFβ, etc.) and pro-angiogenic factors (e.g., VEGF, TNFα, etc.). They enhance differentiation of Tregs, thus promoting tumor growth and sustaining local immunosuppression (cfr. Figure 1). They are mostly distinguished by surface expression of proteins CD163, CD204, and CD206 (Macrophage Mannose Receptor or MMR), but are accompanied by several other receptors (stabilin-1, CD1a, CD1b, CD23, CD93, CD226) [84,85,86,87,88,89]. TAMs may indeed induce carcinogenesis and disease progression as they affect angiogenesis, tissue invasion, and metastasis [85]. Both TAM phenotypes carry the general surface markers CD68, which has been, together with the M2-specific CD163, commonly used for TAM quantification in various solid cancers (breast, colorectal, non-small cell lung, prostate, and ovarian cancer) [84,89]. It seems that high TAM infiltration based on these markers has been correlated to aggressive tumor behavior and increased therapy resistance in breast, ovarian, and prostate cancer [84]. However, differentiating macrophages into anti-tumorigenic M1 and anti-tumorigenic M2 seems to be oversimplified in some cases. Several authors observed the ratio between anti-tumorigenic versus pro-tumorigenic or M1/M2 index TAMs acts as an independent prognostic factor in several malignancies f.i.lung and ovarian cancer [84]. The employed markers used for TAM identification varies as per indication [89,90]. Oral squamous cell cancer (OSCC) in particular shows increased TAM infiltration (CD68^+^, CD11c^+^ or CD163^+^), which was indeed correlated to poor prognostic factors such as increased lympho-vascular invasion and lymph node metastasis [91,92,93]. Furthermore, Weber et al. [94] retrospectively observed that transforming leukoplakic regions (n = 50) had a higher amount of infiltrating TAMs with a significant shift to M2 polarization, based on the M2/M1 ratio via the C163/CD11c index, in comparison to non-transforming leukoplakic regions (n = 53; *p* < 0.001). Similar results were discovered in a population of small OSCC (pT1/pT2, n = 34), in which tumoral samples with lymph node metastasis (N+) showed a higher TAM infiltration and M2/M1 index (based on the IHC staining of CD68, CD11c, CD163, and CD206) than samples from N0 OSCC (*p* = 0.05) [95]. A similar tendency was observed in OSCC and IHC expression of CD206: patients with high infiltration of CD206^+^ cells had more locally advanced disease and lymph node metastasis prognosis [96]. This indicates that the M2 polarization may contribute to malignant transformation and disease progression [94,95,96,97].

A recently published meta-analysis investigated the overall association of TAM to SCCHN based solely on the markers CD68 and CD163, concluding that increased stromal CD168^+^ detection of M2 TAMs was correlated with worse OS. However, the included studies took no account of HPV status in these SCCHN samples [52,90]. Interestingly, recent research from Faustino et al. [97] revealed TAM infiltration (through CD68 and CD163 expression) in OSCC (n = 123) did not affect survival, based on disease-specific survival, but was positively correlated to elevated PD-L1 expression (>10%; clone 22C3, cfr Section 3.1.3, confirming its potential immunosuppressive capacities in SCCHN [98].

TAM assessment could have potential value as a prognostic marker in addition to the well-known TIL evaluation. We observed the TAM assessment in SCCHN is mainly based on IHC expression on FFPE-tissue material while using a restricted set of monoclonal staining antibodies (mostly CD68 and CD163). To make a clear distinction between M1 and M2 TAMs, other markers should be incorporated in future studies.

##### Other Immune Cells

Several other immune cells have been investigated in SCCHN regarding their prognostic and predictive value. These comprise of neutrophils and NK cells (innate immune cells), dendritic cells and MDSC (myeloid cells), and the B cells as lymphocyte subset. Literature regarding these topic remains restricted and is therefore currently of lesser importance as a potential clinical attribute. For a deeper understanding regarding the role of these immune cell subsets, we refer to a recent review from Wondergem et al. [52].

-Neutrophils and NK cells are innate effector cells recruited as first line of defense in case of tissue damage. Although they have been well-described in blood as a marker for inflammation, few data exist regarding their anti-tumor function. Generally, tumor-associated neutrophils or TANs are subdivided in anti-tumorigenic (N1) or pro-tumorigenic (N2) [99]. Two papers viewed high infiltration of polymorphonuclear cells in SCCHN being generally associated with advanced disease, cancer progression, and lower OS [100,101]. NK-cells are lymphocytes that engage in both the innate immunity as an effector cell, and as a regulator of the adaptive immunity due to their IFNγ secretion [102]. Karpathiou et al. [103] investigated NK-infiltration in 152 SCCHN tissue slides using the CD57 protein surface marker. High CD57^+^ cell density in SCCHN was correlated to a lower rate of metastasis and better survival by means of OS and DFS. These CD57^+^ cells were mostly present in OPSCC subsites. Additionally, based on the NK-related transcriptome from the TCGA database, it seems CD56^dim^ marked NK cells are a major part of the immune infiltrate in SCCHN, and has been correlated to increased OS [28]. Reports regarding NK infiltration in SCCHN seems limited; thus, this topic requires further investigation.-Dendritic cells (DC) are myeloid cells functioning as antigen-presenting cells for inducing T-cell activation. General discordance exists about the prognostic role of DC within the immune landscape of SCCHN. Some papers linked higher DC infiltration with positive HPV-status, though this needs to be further elaborated [104]. MDSCs function as suppressors of the native and adaptive immune system. One study depicted MDSC’s are more prevalent in SCCHN-tissue samples than healthy oral tissue via an IHC-based human MDSC marker, MPO [100]. In addition, the number of MPO-stained MDSC also linearly increased according to pathological stage [105]. The value of these myeloid cells, next to TAMs, needs to be further elucidated in SCCHN.-Although B-lymphocytes are key-players in humoral immunity through immunoglobin production, its role in tumor progression is ill-defined. B cells can participate in anti-tumor immunity by enhancing cytotoxic T-cell responses or anti-neoplastic cytokine production, while also being capable of inducing cancer immune evasion [106]. This translates in discrepant findings in literature regarding its prognostic role, for which we refer reader to Wondergem et al. [52]. To this end, we concur that these immune cells are understudied in all types of solid carcinoma.

#### 3.1.3. PD-L1 Status

As stated earlier, the targeting of immune checkpoints, in particular the PD-1/PD-L1 axis, has been a landmark event in the therapeutic field of SCCHN. Anti-PD-1/PD-L1 agents in monotherapy have been implemented as standard first- and second-line treatment in patients with R/M SCCHN [18,53]. Generally, expression of PD-L1 can be observed on tumor cells as well as immune cells, though several studies point out that PD-L1 expression is caused in two different ways. Firstly, PD-L1 expression is temporarily elevated due to T cell induced inflammation. SCCHN is characterized by its high infiltration of TILs, including cytotoxic CD8^+^, which is considered an important prognostic factor [107]. IFNγ secretion from mainly CD8 T cells induces an increased anti-tumor response by stimulating antigen expression and chemokine production (Figure 1), though also induces increased expression of the transmembrane protein PD-L1 as a self-protective mechanism to avoid excessive damage from the inflammation. This generates a pro-carcinogenic environment, as chronic expression of PD-L1 can induce T cell anergy, sustain survival of the transformed cells, thus functioning as a regulator in the dynamic, bidirectional relationship between tumor and TILs [16,73,108]. In this matter, intratumoral CD8^+^ T cells were seemingly inversely associated with PD-L1 expression in OSCC from one retrospective study [109], while Sanchez-Cantelli et al. [110] noticed an increased infiltration of CD8^+^ TILs in HPV^−^ SCCHN with high PD-L1 expression. This finding was associated with better prognosis in terms of disease specific survival (DSS). Then again, these findings were contradicted by other research groups, in which no relationship was found between CD8^+^ T cell infiltration and PD-L1 expression, both studies again performed in retrospective fashion [54,111].

The assessment of PD-L1 expression on a tumor specimen by immunohistochemistry has been acknowledged as a prognostic and predictive biomarker for tailoring use of PD-1/PD-L1 targeting agents in solid cancers, including SCCHN [77]. When treated with ICI, R/M SCCHN patients with elevated PD-L1 expression have higher ORR and increased post-therapy survival compared to low PD-L1 expressing patients [27,58,59,108,112,113]. Still, there are some well-known caveats to be considered when employing PD-L1 expression as a response predictor for ICI that also apply in SCCHN. First, PD-L2 expression is not covered using this PD-L1 assay although PD-L2 has similar function as PD-L1; it interacts with PD-1, induces T cell inactivation and cytokine release, though this is mainly restricted to APC (macrophages, dendritic cells, bone marrow derived mast cells, and peritoneal B cells). Subsequently, high PD-L2 expression is associated with promotion of metastasis and poor prognosis (DFS, DSS) in various solid cancers [114,115,116]. Second, patients with PD-L1 overexpression do not necessarily respond to ICI, while patients with limited or even absent PD-L1 expression did respond to ICI [117]. Third, evaluation of PD-L1 expression has been technically challenging due to the implementation of different (trial-validated) PD-L1 assays depending on the pharmacy-developed ICI agent, while also employing different types of platforms used for the immunohistochemical staining. In addition, the PD-L1 evaluation method for every immunostaining antibody has their own threshold and cut-off value for defining PD-L1 positivity (Table 1).

Fourth, a significant degree of intratumoral heterogeneity exists in regard to tumor cell staining: PD-L1 positive cells tend to be located at the periphery of the tumor adjacent to the peritumoral stroma. Immune cell staining has been noted to be similarly heterogeneous: immune cell staining seems primarily localized to the peritumoral stroma with few TILs in most cases. This pattern of cuffing of tumor by a PD-L1 positive immune cell infiltrate has been noted to be particularly characteristic of HPV-driven oropharyngeal carcinoma [118]. This may be a possible explanation for the underscoring of aspirate cell blocks and core biopsies relative to resected specimens and excisional biopsies is a failure to sample the peritumoral stroma in small biopsies reported in several articles [119,120,121]. Lastly, PD-L1 expression is subjected to temporal heterogeneity: immune and tumor cells are continuously shifting shape and functionality during cancer development and progression. An assessment of PD-L1 expression is only one specific time-frame and insufficiently apprehends the dynamic evolution TME of the tumor, making PD-L1 expression a time-dependent biomarker [122].

Altogether, the scoring and interpretation of PD-L1 expression has been made complex. It is without question that PD-L1 expression is an interesting player in the field to assess the TME and tailoring use of ICI, but this biomarker lacks robustness, and its assessment is complicated due to several technical issues. As mentioned earlier, there remains controversy regarding the potential prevalence of discrepancies in PD-L1 scoring due to interspecimen and temporal heterogeneity (biopsy, resection, or lymph node or organ metastasis). To our knowledge, intratumoral heterogeneity has only been assessed in SCCHN by two studies: both evaluated PD-L1 expression by means of the Combined Positivity Score (CPS) in biopsy material versus the matching resection material. Results showed significant discordance in material with absent PD-L1 expression, confirming the risk of underscoring in biopsy specimen [121,125]. Two studies investigated PD-L1 expression on primary SCCHN versus associated (non-recurrent) lymph nodes and found a clear correlation in expression pattern [125,126]. Nonetheless, literature remains scanty in this domain: further specimen-subsite analysis is required and aforementioned results should be validated by additional studies. Exploring the association between PD-L1 expression, HPV-status, and other TME-associated biomarkers (TILs and TAMs) should be elucidated.

### 3.2. Genetic Biomarkers

#### 3.2.1. Micro-Satellite Instability (MSI)

The DNA mismatch repair (MMR) system is an indicator for genomic (in)stability as DNA repair gene products. It makes use of different enzymatic protein complexes (MutS, MuH, and MutL) to bind at damaged DNA, repairing base pair mismatches or insertion/deletion loops. Afflicted genetic damage (through transmitted germline mutations or epigenetic changes) to this DNA MMR system or loss of function may eventually lead to formation of MSI. Indeed, several types of solid carcinoma, mostly subsets of colorectal cancer, have been associated with MSI via dysfunctional expression of several MMR proteins (formed as proteins complexes such as MSH2-6, MSH2-3, PMS1-MLH1, MLH1-3) [127]. Subsequently, MSI-high tumors may be correlated with a higher tumor mutational burden (TMB), and thus, an increased production of neo-antigens, leading to higher T cell reactivity. From this perspective, MSI-high tumors may have better ORR towards treatment with ICI. In SCCHN, MSI has not been implemented in routine testing [128]. The prevalence of MSI positivity in SCCHN has been estimated at 1%–26% [129,130,131,132]. The prognostic value of MSI in SCCHN remains unclear due to low sample-size and few studies, thus lacking statistical power. Initially, it seemed that MSI positivity in SCCHN was correlated with a higher probability of local recurrence in tissue specimen with negative resection margins [133]. A recent study investigated the role of MMR deficiency as a response predictor for anti PD-1 agents in twelve different solid carcinoma, though SCCHN was not included. Nonetheless, a radiological ORR was observed in 53% of patients, of which almost half of them reaching a complete response [134]. An interesting report described a patient with metastasized SCCHN, low PD-L1 expression, and high MSI, who was submitted to a PD-L1 inhibitor. This patient experienced a durable and complete response [135]. The role of MSI as a potential biomarker for tailoring ICI use in SCCHN has not been thoroughly investigated as a predictor of ICI-responsiveness. Current reports indicate that incidence of MSI is fairly low in SCCHN, which reduces its potential implementation as a routine diagnostic test in SCCHN. Nevertheless, its role should be elaborated in future clinical research.

#### 3.2.2. Genetic Screening


*Type of material*


Genetic-based prognostication and classification have been a major part in the unravelling of the immune contexture of SCCHN. Classically, tissue specimen are utilized for DNA/RNA extraction and genetic analysis using next-generation sequencing techniques (NGS) to identify the mutational landscape. Nonetheless, (repetitive) tissue biopsies are an invasive and costly procedure, which has increased the popularity for applying collection of liquid biopsies. The introduction of cell-free (cf) DNA and circulating tumor (ct) DNA testing have been welcomed as innovative techniques of genetically profiling solid carcinoma. Furthermore, electronic platforms such as the TCGA database and cBioPortal displays the patterns of gene expression and clinical data from several clinical trials in (solid) malignancies. This allows universal access to genetic information, and thus can function as useful tools in the development of a genetic prognosticator for (future) prospective trials [136,137].


*Tumor mutational burden (TMB)*


The TMB can be utilized to quantify the neoantigen load via the number of (non-synonymous) mutations present in a specific tumor (mut/mB) [20,127,138]. TMB can be acquired through whole-exome or whole-genome DNA sequencing (mostly done on tumoral tissue material). In non-squamous cell lung cancer, patients with higher TMB were correlated to significant clinical benefit when treated with ICI [139,140]. SCCHN is considered one of the (tobacco-related) solid malignancies with the highest TMB, around 5-10mut/MB [20]. Several clinical trials confirmed that presence of high TMB was correlated to better prognosis when treated with ICI [137,138,141,142]. However, this was mostly in case of virus-negative (HPV^−^/EBV^−^) SCCHN, which have higher TMB due to prevalence of secondary risk factors (smoking/alcohol consumption). Furthermore, TMB seems to be independent of PD-L1 expression and gene expression profiles in the tumor, confirming that the TME is a very dynamic but complex entity [142]. There is high interest in integrating TMB as a prognostic or predictive biomarker in patients with SCCHN, especially before initiating treatment with immunomodulating agents (f.i. ICI), but this should be further investigated in future prospective trials.


*Gene signatures*


In SCCHN, several genes have been correlated to carcinogenesis and the generation and modulation of anti-tumor immune response. As mentioned earlier, SCCHN is genetically heterogeneous, which has complicated the search for an evident immune-related gene signature. Multiple studies investigated different sets of immune genes signatures to predict prognosis in the population while also tailoring the use of immunotherapy through assigning different immune-gene based subgroups to various immunotherapy based strategies [7,140,143,144,145,146,147,148,149,150,151,152,153,154,155,156]. Results from these studies have generated an immune-related gene panel or signature based on 10 to 27 different genes (cfr. Table 2), which were mostly DNA and/or RNA based (mRNA, lncRNA, miRNA). An extensive elaboration for each acquired gene signature falls beyond the scope of this paper. Nonetheless, these genetic signatures might be insightful in regard to their predictive and prognostic value, especially when considering treatment with ICI.

Again, before clinical implementation of the gene signature could take place, uniformity should exist regarding the in-and exclusion of relevant genes and the type of material used for genetic screening (primary tumor versus metastasis, tissue versus liquid biopsy). Integration of a gene signature should go hand in hand with an extensive examination of the abovementioned TME-associated biomarkers. In this regard, SCCHN may be subdivided based on their immunogenic properties into low, moderate, or highly inflamed phenotypes [12]. An interesting approach would be the construction of immune-related nomogram or immunogram in SCCHN providing important prognostic and predictive with the potential of tailoring use of immunotherapeutic agents [148].

### 3.3. Circulating Blood Cells

As systemic inflammatory responses may be reflected using blood parameters, some of them may be beneficial as predictive and prognostic biomarkers for minimal invasive and clinical monitoring of patients. Several peripheral blood parameters have been investigated and explored: absolute lymphocyte count (ALC), absolute neutrophil count (ANC), absolute eosinophil count (AEC), absolute monocyte count (AMC), or neutrophil to lymphocyte ratio (NLR) being the most noteworthy parameters. However, literature describes various discrepancies of these parameters between studies, thus its clinical value is yet to be portrayed [156,157,158,159,160,161,162]. Interest has therefore shifted to the characterization of immune and non-immune-inflamed phenotypes using flow cytometric analysis (FCA). This is a common and widely available technical approach to quantify and identify immune cell profiles using cell surface-expressed markers in blood and blood derivates [163]. As mentioned in Section 3.1.2, SCCHN is known to have an immunosuppressive environment with lower absolute T cell counts, in particular CD4^+^ and CD8^+^ T cells, and dysfunctional CD8^+^ effector cells. On the other hand, immunosuppressive Tregs and MDSCs are elevated [162,164,165].

Indeed, unravelling the immune-profile in SCCHN of circulatory blood cells is promising and may go hand in hand with abovementioned (tissue-based) parameters. For instance, identifying and quantifying the repertoire of different T cell subsets by FCA, of which CD4^+^ and CD8^+^ T cells are the main players on the pitch, could be accompanied by an IHC-based TIL subtyping in tumoral tissue. Below, we describe some currently known dysregulations in blood and blood-derivates observed via FCA detected surface markers in SCCHN.

As mentioned earlier, an important subset of CD4^+^ T cells in solid carcinoma are Tregs. These are grossly divided into three main groups according to the surface marker-based phenotype and their function: (1) immunosuppressive resting (CD45RA^+^ FoxP3^low^) Tregs; (2) activating (CD45RA^−^ FoxP3^high^) Tregs; and (3) non-immunosuppressive (CD45RA^−^ FoxP3^low^) T cells [166].

Effector CD8^+^ T cells on the other hand, should be divided into naïve T cells, central memory T cells, effector memory T cells, and effector T cells, the latter being responsible for cancer cell elimination after antigen stimulation [165].

A prospective study analyzed base-line T-cell populations by FCA on peripheral blood of 85 oropharyngeal and laryngeal cancer patients compared to healthy controls. A shift was noted from naïve CD8^+^ T cells into effector memory T cells, while the total amount of effector T cells and effector memory T cells was elevated in HPV^+^ in comparison to HPV^−^ OPSCC [167].

A preliminary prospective study from Boucek et al. [168] compared the Treg frequency (determined as CD4^+^CD25^+^ Tcells) in the blood of SCCHN patients (n = 112) at the time of diagnosis compared to healthy donors. A general increase in Tregs in the SCCHN group was observed. Furthermore, a significantly higher level of circulatory Tregs was associated with recurrent disease for SCCHN.

Lechner et al. [54] observed Tregs (identified as CD4^+^/FoxP3^+^ CD127^low^) in blood (PBMC’s) to be elevated in the SCCHN group compared to healthy donors. These results were equally confirmed on paired tissue samples. Subsequently, a decrease in circulating CD4^+^ naïve T cells (CD4^+^/CD45RO^−^/CD27^+^) and an increase of memory T cells (CD4^+^/CD45RO^+^) were observed in comparison to control groups. Of notice, CD45RO^+^ is a glycoprotein that represents the prevalence of tumor-infiltrating memory T lymphocytes, and thus the activation status of T cells in general [169].

Attention has also been brought to the migratory capacity of circulating CD4^+^ and CD8^+^ T cells by Andrade et al. [170]. Based on the expression of surface proteins CD18, CD54, and CD62L, significant alterations were noted on CD4^+^ T cells and particularly, CD8^+^ T cells. These migratory lymphocytes could be an essential component for the local and/or systemic inflammatory response during carcinogenesis, but this needs to be further elaborated. Furthermore, a reduced number of circulating CD38^+^/CD8^+^ T cells was observed in patients with highly advanced/metastatic SCCHN, CD38 being a both sensitive and specific marker for immune activation in various diseases. Again, its clinical role needs to be defined in future research [170].

Circulating NK cells only provide little data regarding their role in SCCHN. It seems the total amount of (CD16/56^+^) NK cells is substantially lower compared to control groups. Low amounts of invariant NK Tcells were correlated to poor outcome in SCCHN [168,171]. Essentially, aforementioned authors concur the phenotypic analysis of circulating lymphocytes could be interesting for assessing the immunological status, monitoring the clinical course, or as prognostic and predictive tools for patients with SCCHN.

As mentioned in Section 3.1, MDSC are immature immune cells known for negatively regulating immune responses in pathological conditions, including carcinogenesis. MDSCs comprise of different subsets expressing various myeloid markers such as CD11b, CD33, CD14, CD15, and CD16, while lacking the expression of HLA-DR. These surface markers are crucial for identification by FCA. Several reports indicate CD14^+^ HLA-DR^−^ cells induce immunosuppression by inhibiting T cell proliferation through augmented expression of immune checkpoint molecules, increased release of cytokines (TGFβ, IFN-γ) and enzymes (ROS, NOS, Arg1) [41,172]. Of course, these mechanisms differ according to the type of solid carcinoma. In SCCHN, (CD14^+^ HLA-DR^-^) MDSCs obtain their immunosuppressive effects through upregulated expression of PD-L1 and an increased release of TGFβ [173]. Regarding clinical significance, it seems the amount of circulating MDSCs quantified by FCA is substantially lower in peripheral blood samples of SCCHN and concentrations are correlated to the tumoral burden in SCCHN, which is in concordance with abovementioned tissue-related results [174,175,176]. Hereby, MDSCs may indeed be of biological importance during tumor progression.

Lastly, the detection of immune checkpoint molecules on circulating T cells via FCA has also been introduced. Immune checkpoints (f.i. PD-1) expressed on T cells have low sensitivity using immunohistochemical detection on tumoral tissue, hence the recommendation of combining both techniques in the future for adequately assessing the PD-L1 status of SCCHN [54,177]. In summary, aforementioned data about circulating blood cells and their expressed surface proteins seem worthwhile for contemporary evaluation, providing further insight in tumor biology and the immunological profile of SCCHN.

### 3.4. Oral Microbiota

Microbiota have an indisputable role in the development and maturation of the host immune system. Mounting evidence has correlated its dysregulation to several health issues, the most common being systemic auto-immune, inflammatory bowel, cardiovascular, and metabolic diseases [178,179]. In addition, the immunomodulatory role of gut microbiota has been extensively described in malignancies, it being associated with immune dysregulation, disease progression, and as a regulator of immunomodulotary therapeutics such as ICI [180,181,182]. As of today, no data have linked squamous cell cancer in the head and neck to gut microbiota. However, exciting evidence has been presented concerning oral microbiota. The earliest paper found significant discrepancies in oral bacteria compositions between healthy and OSCC subjects, the latter being colonized species and strains that induce chronic inflammation, hence potentially inducing carcinogenesis and cancer progression. Furthermore, smoking and excessive alcohol abuse, two major risk factors for SCCHN, seem to reduce bacterial heterogeneity [183]. To this end, Shin et al. [184] observed an increased colonization of *Lactobacilli* while *Haemopilus, Neisseria, Gemellaceae, or Aggregatibacter* were downregulated. Recently, it was shown that enrichment of the *Fusobacterium*, *F. nucleatum* in SCCHN patients showed lower tumor stage, lower rate of recurrence, and lymph node metastasis. Generally, a shift seems to occur (in SCCHN compared to healthy tissue) in which *Streptococci* seem to be replaced by *Fusobacterial* strains [184]. Although the clinical significance of microbiota in regard to therapeutics within SCCHN needs to be further elaborated, a respectable attempt has been made by Ferris et al. [185]. As a sub-investigation of the CHECKMATE 141-trial, the saliva of 82 ICI-treated R/M SCCHN patients were assessed for distinctions in bacterial colonization by profiling using high-throughput 16S ribosomal RNA sequencing, and compared to healthy controls. Although no prognostic nor predictive correlation could be made (partially explained due to the limited sample size and low ORR), these are welcoming attempts that should incite further investigation [20,185].

## 4. General Conclusions and Future Perspectives

As ICI have found their way in the standardized practice of oncology medicine, there is a necessity of obtaining a better understanding regarding the TME to further personalize ICI treatment. We know that there is an indisputable correlation between the TME and the recognition and elimination of neoplastic cells. Further deciphering of the TME may thrive discovery of potential diagnostic, prognostic biomarkers or additional targets for enhancing the effectiveness of ICI, as ICI-resistant clones have also been observed in cancer patients [146]. In SCCHN region, there is no denying that the TME is of major importance regarding prognosis and response prediction to ICI. Until today, only p16/HPV status and PD-L1 expression have been integrated in routine clinical practice as clinicians and scientists agree that these biomarkers harbor important prognostic value [108]. Identification and quantification of the immune infiltrate, in particular TIL subsets, have been thoroughly investigated these last decades in SCCHN. However, due to its great inconsistencies regarding predictive and prognostic information, no consensus has been reached for designing a universal methodology in TIL scoring, nor integrating TILs in the diagnostic landscape of SCCHN. Although some potential may lie in the evaluation of TAMs in the TME, there is still a lack of solid scientific evidence in SCCHN; thus, further exploration is warranted. As ICI have been integrated in the therapeutic landscape of cancer, the clinical value of PD-L1 expression in SCCHN has been skyrocketing. However, next to the well-known technical issues (different methodologies, immunostainers, and platforms), temporal, and intra- and intertumoral heterogeneity do further complicate the assessment. Currently, PD-L1 expression lacks specificity and robustness to be employed as a sole predictor of ICI-response in SCCHN. We are in desperate need of additional biomarkers to adequately profile the TME in SCCHN and tailor the use of immune checkpoint inhibitors.

When considering genetic-based biomarkers, TMB and MSI may be valuable predictors of response to immunomodulating agents, but fall short due to the low prevalence in this population. Gene signatures hold more potential for a personalized strategy, though current studies lack uniformity on which SCCHN-and/or immune-related genes should be targeted for screening. The same accounts for circulating blood biomarkers, holding prognostic or predictive information but lacks uniformity in methodology and, consequently, in results. Lastly, as the microbiome has recently emerged as a popular mediating factor of immunology and cancer, the oral microbiota and its link to SCCHN could be another path worth exploring for additional biomarkers.

In general, SCCHN prognosis prediction should not only be based on the clinical staging system (TNM or AJCC), differentiation grade, and HPV and PD-L1 status. Ideally, future studies should partially integrate above described biomarkers to construct a unanimous TME-based classification that includes the assessment of tissue-infiltrating and circulating immune cells and gene signatures. An interesting approach would be the construction of an immunogram as suggested by Blank et al. [148], assigning SCCHN into different immunoprofiles based on their immunogenic properties. SCCHN remains a complex and heterogeneous cancer. Several biomarkers have already been investigated, demonstrating their worth in SCCHN, but are prone to flaws due to retrospective study designs, lack in uniformity of assessment, no standardization, and discordancy in published results. Therefore, improving our understanding of the TME and the dynamic tumor-immune cell interactions will be crucial to explore, identify, develop, and ultimately integrate new prognostic and predictive biomarkers into clinical care.

## Figures and Tables

**Figure 1 cancers-13-01714-f001:**
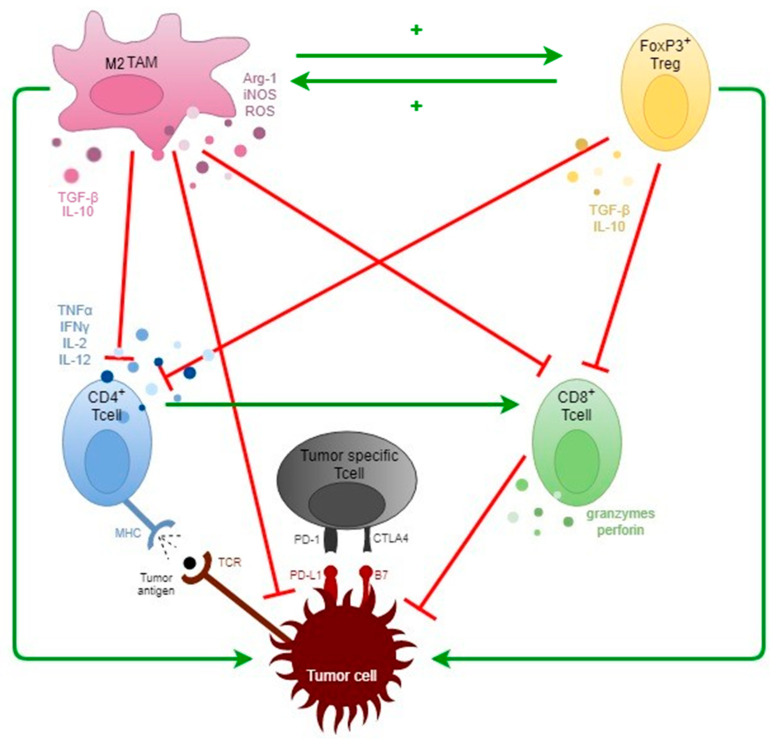
A simplified overview of interactions between tumor cells and TME-associated immune cells. CD4^+^(FoxP3^−^) and CD8^+^ T cells are designed for tumor cell elimination, though their function can be altered by recruitment of pro-tumorigenic (M2) TAMs and FoxP3^+^ Treg, who interact in a positive feedback-loop. TNFα = Tumor necrosis factor alfa. TGFβ = Tumor growth factor Beta; IL = Interleukin; IFNγ = Interferon gamma; Arg1 = Arginase 1; iNOS = inducible nitric oxide synthase; ROS = reactive oxygen species, PD-1 = programmed death 1, PD-L1 = programmed death ligand 1, CTLA-4 = cytoxic T-lymphocyte associated protein 4, MHC = major histocompatibility complex, TCR = T cell receptor.

**Figure 2 cancers-13-01714-f002:**
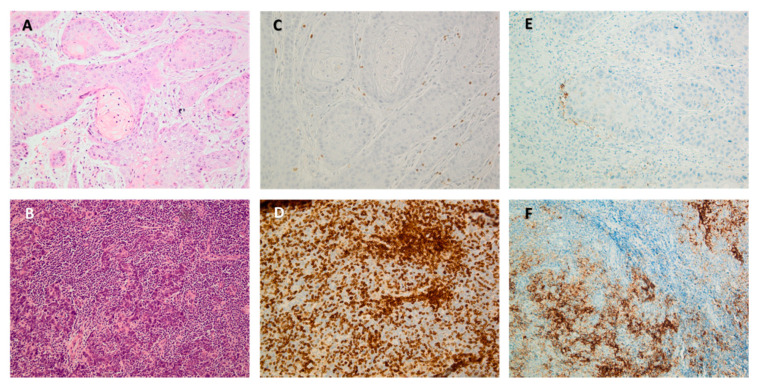
Stained squamous cell cancer of the head and neck (SCCHN) sections. (**A**,**B**) comparison of low versus high stromal tumor infiltrating lymphocyte (TIL) presence, stained with haematoxilin-eosin; (**C**,**D**) comparison of low versus high stromal CD3^+^ T cells, stained with CD3 primary antibody clone F7.2.38 (Dako, Glostrup, Denmark); and (**E**,**F**) comparison of low versus high PD-L1 expressing tumor and/or immune cells, stained with the PD-L1 clone 22C3 (Agilent-Dako, United States). **A** + **C** and **B** + **D** represent sections from the same patients.

**Table 1 cancers-13-01714-t001:** Different antibodies and platforms for ICI tested in SCCHN.

ICI Agent	Complement PD-L1 Ab	Ab Host Species	Platform	Detection System	Diagnostic Cut-Off
Pembrolizumab [59,112]	22C3	murine	Dako autostainer Link 48	EnVision FLEX visualization system	TC or IC ≥1% (CPS)
Nivolumab [18]	28-8	rabbit	Dako Autostainer Link 48	OptiView DAB IHC Detection Kit	TC >1%, TC >5%
Atezolizumab [123]	SP142	rabbit	Ventana Benchmark Ultra	OptiView DAB IHC Detection Kit	TC: ≥ 5%, IC: ≥ 5%
Durvalumab [60]	SP263	rabbit	Ventana Benchmark Ultra	OptiView DAB IHC Detection Kit/ OptiView Amplification Kit	TC: ≥ 25%
Avelumab [124]	73-10	rabbit	Dako autostainer Link 48	OptiView DAB IHC Detection Kit	N/A

Ab = antibody; CPS = combined positive score; IC = immune cells; ICI = immune checkpoint inhibition; IHC = immunohistochemistry; N/A = not applicable; SCCHN = squamous cell carcinoma of the head and neck; TC = tumor cells.

**Table 2 cancers-13-01714-t002:** Overview of some potential gene signatures in SCCHN.

Study Reference	Associated Genes
[7]	AJUBA (−), CASP8 (−), CD56 (+), CD8 (+), CDKN2A (−), EGFR (−), FAT1 (−), FGFR2 (+), HRAS (−), LAG3 (+), NOTCH1 (−/+), PIK3CA (−/+), TP53 (+), TP63 (−/+), TRAF3 (+)
[140]	CCL5, CD27, CD274, CD276, CD8a, CMKLR1, CXCL9, CXCR6, HLA-DOA, HLA-DRB1, HLA-E, IDO1, LAG3, NKG7, PDCD1GL2, PSMB10, STAT1, TIGIT
[143]	AVPR2, BTC, CCL22, CCR6, CHGB, DKK1, HBEGF, HRG, ICOS, IL20RA, INHBB, KLRK1, LCNL1, MASP1, OLR1, PDGFA, PTX3, RBP4, RFXAP, ROBO1, RORB, SH3BP2, TMSB4Y, TNFRSF4, TNFRSF18, TNFRSF25, ULBP1
[147]	BATF, CCL11, CCR4, CCR7, CD27, CD79B, CMA1, CNR2, CTLA4, CTSG, GZMM, IL16, IL19, MASP1, PGLYRP4, SAA1, TNFAIP3, TREML1
[149]	AJUBA (−), CASP8 (−), CCND1 (−), CDKN2A (−), EGFR (−/+), FAT1 (−), FGFR1 (−), FGFR3 (+), HLA-A (−/+), HRAS (−), KMT2D (−), MYC (−), NOTCH1 (−/+), NSD1 (−), PIK3CA (−/+), TP53 (−), TP63 (−/+), TRAF3 (+)
[150]	CDKN2A (−), CUL3 (−), FGFR3 (+), FLG (−/+), MLL2 (−/+), MLL3 (+), NOTCH1 (−/+), NOTCH2 (−), NSD1 (−), PIK3CA (−/+), TP53 (−), UBR5 (−)
[151]	AJUBA (−), B2M (+), CCND1 (−), CDK4 (−), CDK6 (−), CDKN2A (−), CUL3 (−), E2F1 (+), FAT1 (−), FGFR2 (+), FGFR3 (+), HLA (+), HRAS (−), KEAP1 (−), KRAS, NF1, NF1 (+), NFE2L2 (−), NOTCH1 (−/+), NRAS, PIK3CA (−/+), RB1 (−), TP53 (−), TP63 (−/+), TRAF3 (+)
[152]	GZMA (+), GZMB (+), IDO1 (+), IFNG (+), LAG3 (+), PRF1 (+)
[153]	CYLD (+), EP300 (+), FGFR3 (+), KMT2D (+), NFE2L2 (+), PEG3 (+), PIK3CA (+), RB1 (+), STAT3 (+), TSC2 (+)
[154]	ADGRV1 (−), CCND1, CDKN2A (−), CDKN2B (−), EGFR (−), FAT1 (−), FAT2 (−), FAT4 (−), KMT2C (−/+), KMT2D (−), NFE2L2 (−), NOTCH1 (−), PIK3CA (−/+), RELN (−), TP53 (−)
[155]	AKNA, ARHGAP9, CCR7, CORO1A, GIMAP4, GIMAP7, IL10RA, ITGAL, ITK, P2RY8, PPP1R16B, PRKCB, SASH3, SP140, TBC1D10C, TRAF3IP3

All genes are ordered alphabetically per study. (−) = common mutation in HPV negative tumors; (−/+) = common mutation irrespective of HPV status; (+) = common mutation in HPV positive tumors; SCCHN = squamous cell carcinoma of the head and neck.

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
