# Peer review of "Immuno-Oncological Biomarkers for Squamous Cell Cancer of the Head and Neck: Current State of the Art and Future Perspectives"

_cancers, 2021, doi:10.3390/cancers13071714_

Round 1
Reviewer 1 Report
This review provides the available results on emerging immune predictive biomarkers in HNSCC with comments on the therapies that could be developed.
Although in the scientific literature there are reviews with a fairly similar topic, this manuscript has the advantage of being well written, well organized and obviously it comments on the most recent research articles published.
The manuscript is suitable for the publication, however a very interesting topic such as the oral microbiota is recently becoming a very interesting topic in oral diseases including also oral-pharyngeal and laryngeal tumors (ie: doi: 10.3390 / ijms21207621, doi: 10.3390 / cancers12113425., https://doi.org/10.1093/annonc/mdy507).
For completeness and to increase the novelty of the manuscript, the authors could comment on this topic as potential biomarkers in HNSCC Immunotherapy.
Reviewer 2 Report
The present review is carefully presented, however there is not enough clarifications. Several comments should be addressed.
1) In Introduction section authors should emphasize the crucial role of immune system in cancer progression.
2) In general, clarifications are required - both for biomakers and for methods of studying these biomarkers, as well as a description of the patient cohorts, experimental models, types of tumors.
All of these issues have to be identified.
3) Figure 1 – it is not clear, what do arrows up and down mean?
M2 macrophages also express and secrete Il-10 and TGFb. It should be indicated in Figure.
What about the interaction between FoxP3 and M2 TAMs?
4) Paragraph 3 – the authors should clearly distinguish whether they are talking about biomarkers or cells of the tumor microenvironment. Otherwise it is wrong to insert both a description of cells and a description of biomarkers in one paragraph in distinct sections.
3) Paragraph 3. The TME in SCCHN –What are the specific features of the tumor microenvironment in SCCHN? Which cells are the majority? If there are such differences compared to other cancer localizations, it is better to mention them.
4) page 8 “….Authors agreed that the ratio pro-tumorigenic versus anti-tumorigenic (M1 / M2) has better potential as an independent prognostic factor in malignancies...” There is a mistake – have to be M2/M1.
The ratio M2/M1 has prognostic value not in all cancers. It is necessary to indicate the cases and types of tumors where it matters.
In which cancers M2 macrophages have prognostic value? Examples should be addressed in more detailed manner. The following articles can be helpful: https://www.frontiersin.org/articles/10.3389/fonc.2020.566511/full, https://pubmed.ncbi.nlm.nih.gov/24507759/, https://pubmed.ncbi.nlm.nih.gov/31182919/, https://pubmed.ncbi.nlm.nih.gov/29361917/, https://pubmed.ncbi.nlm.nih.gov/30089583/, https://pubmed.ncbi.nlm.nih.gov/32208142/
CD68 is a general marker of macrophage population, it is not M1 marker.
For M2/M1 ration and for M1, M2 populations all markers (CD163, CD206, CD204m HLA-DR, iNOS etc.) have to be addressed.
The sentence “….M2-polarized TAMs have also been correlated to CD206 expression, a macrophage mannose receptor….” is improperly, because CD206 is a marker of M2 TAMs, and M2-polarized TAMs cannot be correlated to CD206 expression. Or authors have to indicate which marker was used for M2?
5) The title of this article “The immunological landscape in squamous cell cancer of the 2 head and neck: current biomarkers and future perspectives” – however authors also describe genetic biomarkers in section 3.2. This genetic alterations occur in tumor cells or in immune cells? If they refer to tumor cells – the title should be renamed or this section (3.2) should be deleted.
6) Section 3.3 Circulating blood - Are authors talking about circulating biomarkers regardless of their expression on immune cells or strongly expressed on them? Please indicate patient cohorts and the type of material in referenced studies.
7) The title of this article “The immunological landscape in squamous cell cancer of the 2 head and neck: current biomarkers and future perspectives” implies that the role of all cells of the immune system will be discussed in the progression of SCCHN , but the review displays the characterization only of TAMs and T cells. What is about NK cells, neutrophils, B cell?
8) Reference list should be corrected according to the journal requirements.
Reviewer 3 Report
This is a very "compact" review of the "immunological landscape in SCCHN", still it covers more, so the title should be a bit modified.
On page 11 the shortening TMB must be explained.
Round 2
Reviewer 2 Report
The authors have done a great job. The manuscript can be accepted. However, there are several mistakes throughout the text, so English spelling and text check are required for the final acceptance.